# Acute Gaseous Air Pollution Exposure and Hospitalizations for Acute Ischemic Stroke: A Time-Series Analysis in Tianjin, China

**DOI:** 10.3390/ijerph192013344

**Published:** 2022-10-16

**Authors:** Mingrui Cui, Changqing Zhan, Wenjuan Wu, Dandan Guo, Yijun Song

**Affiliations:** 1General Medicine Department, Tianjin Medical University General Hospital, Tianjin 300052, China; 2Department of Neurology, Wuhu No.2 People’s Hospital, Wuhu 241001, China

**Keywords:** air pollution, ischemic stroke, hospitalization, gaseous pollutant

## Abstract

Background: Stroke has always been an important problem troubling human health. Short-term exposure to air pollutants is associated with increased hospital admissions. The rise of pollutants such as O_3_ has caused a huge social and economic burden. This study aims to explore the relationship between short-term exposure to ambient gaseous pollutants and daily hospitalizations for ischemic stroke, utilizing a four-year time-series study in Tianjin. Methods: Collecting the data of gaseous pollutants (NO_2_, SO_2_, CO, O_3_), meteorological data (including daily average temperature and relative humidity) and the number of hospitalizations due to ischemic stroke in Tianjin Medical University General Hospital from 2013 to 2016. Poisson regression generalized additive model with single-day and multi-day moving average lag structure was used to estimate adverse effects of gaseous pollutants on daily hospitalizations. Subgroup analysis was performed to detect modification effect by gender and age. Results: In total, there were 9081 ischemic stroke hospitalizations. After controlling for the meteorological factors in the same period, no significant findings were found with the increase of NO_2_, SO_2_, CO and O_3_ concentrations at most of the time in the single-pollutant model. Similarly, in the stratified analysis, no associations between gaseous pollutants and ischemic stroke were observed in this study. Conclusions: Short-term exposure to NO_2_, SO_2_, CO and O_3_ was not distinctly associated with daily hospitalizations for ischemic stroke in Tianjin. Multicenter studies in the future are warranted to explore the associations between gaseous pollution exposure and ischemic stroke.

## 1. Introduction

Stroke has always been an important problem troubling human health. The World Health Organization notes that stroke ranked second in the global cause of death statistics in 2016, and the more economically developed a country is, the more dominant stroke is in the proportion of causes of death. In China, stroke is the leading cause of death [1], resulting in a huge social and economic burden. Some studies outside of China have shown that exposure to air pollutants such as NO_2_, SO_2_, CO and O_3_ increases the risk of cerebrovascular disease, hospitalization and even death [2,3,4,5,6,7,8]. Surveys in China have also shown short-term exposure to air pollutants was associated with increased hospital admissions for coronary heart disease, and the total medical cost in Sichuan Province that could be attributable to exceeding PM_10_ and PM_2.5_ were 42.04 and 67.25 million CNY from 2017 to 2018, respectively [9]. Currently, environmental pollution in China has reached an urgent situation. Tianjin, located in the North China Plain, is an important location for industry and traffic in the north of the country. The air pollution caused by chemical smoke emissions, automobile exhaust, coal combustion and so on is serious. With the vigorous control of environmental pollution in recent years, the air quality in Tianjin has improved; however, there is still an overall upward trend of pollutants such as O_3_. Therefore, the purpose of this study is to explore the effects of short-term exposure to gaseous pollutants NO_2_, SO_2_, CO and O_3_ on the number of patients with ischemic stroke in Tianjin. Furthermore, this study aims to provide a theoretical basis for the formulation of more targeted strategies for the prevention of ischemic stroke.

## 2. Materials and Methods

### 2.1. Clinical Data

The admission data of Tianjin Medical University General Hospital from 1 January 2013 to 31 December 2016 were collected, including patient name, age, gender, diagnosis, admission time, home address, work unit and so on. The subjects of this study were the patients age over 18 years of age. The diagnosis of the patient was based on the discharge diagnosis, referring to the International Classification of Diseases (ICD-10), and the study selected the disease as ischemic stroke (I63.0–I63.9).

### 2.2. Environmental Data

The meteorological data came from China Meteorological data Service Network (http://data.cma.cn/, accessed on 11 October 2022), including daily average temperature (temperature, temp, °C) and relative humidity (relative humidity, rh, %). The automatic observation project of the weather station conducts 24 regular observations a day and takes the average value. Real-time hourly air quality data (PM_2.5_, PM_10_, NO_2_, O_3_, SO_2_ and CO) of Tianjin city from 1 January 2013 to 31 December 2016 were retrieved from Tianjin Meteorological Bureau (26 observation stations were set up in the entire city, including National Meteorological Observation Station: Beichenkejiyuanqu: 25,576, Binshuixidao: 3142, Dazhigubahaolu: 14,686, Disidajie: 14,640, Donglizhongxue: 10,984, Hanbeilu: 14,254, Hangtianlu: 14,694, Hedongzhan: 10,987, Hexiyijinglu: 3142, Hexizhan: 10,986, Jichecheliangchang: 10,988, Jidianqichang: 10,987, Jingulu: 3142, Konggangwuliujiagongqu: 10,995, Meijiangxiaoqu: 8238, Nanjinglu: 20,566, Nankoulu: 13,328, Qianjinlu: 14,674, Qinjianlu: 14,676, Shijiancezhongxin: 12,895, Taifenggongyeyuan: 10,994, Tianshanlu: 12,894, Tuanbowa: 25,322, Yongminglu: 25,559, Yuejinlu: 14,535, Zhongxinshengtaicheng: 10,995) and Tianjin Eco-Environmental Monitoring Center; the unit of CO concentration was mg/m^3^, and the rest were μg/m^3^. Data were detected by ground monitoring station. The missing data were supplemented by the interpolation method (Appendix A). Data validation was completed by Tianjin Meteorological Bureau with reference to Quality Control of Surface Meteorological Observation Data.

### 2.3. Statistical Methods

Poisson distribution is suitable for dealing with a situation of low probability of occurrence but many instances of repetition, and is often used to study the influence of exposure factors on the results. In the time series analysis, patient admission was a small probability event, while the population base was huge, and the number of admissions per day conformed to Poisson distribution, so this study used the generalized additive model Poisson distribution for data processing. In this study, the generalized additive model of Poisson distribution in time series analysis method was used to analyze the relationship between the concentration of atmospheric pollutants NO_2_, SO_2_, CO, O_3_ and the daily admission number for stroke, and to control the possible related factors such as temperature, humidity, holidays and other factors to reduce the interference with regard to the analysis results. For the establishment of a specific model, we created the formula: lgE(Yt) = α + β × C Pollutant, lag + ns (time, df) + ns (temperature, df) + ns (humidity, df) + DOW + Holiday.

In the above formula, “Yt” represents the number of patients admitted for cerebral infarction on the observation day; “E (Yt)” represents the expected value of the number of patients admitted for cerebral infarction during the t days; “α” is the intercept; “β” is the regression coefficient; “C Pollutant, lag” is the pollutant concentration when the observed lag days; “ns” is a natural cubic spline smoothing function; “df” is the degree of freedom; “DOW” is the week dumb element variable; “Holiday” is the festive dummy element variable; “time” is time; “temperature” is the average temperature; “humidity” is the relative humidity. The degree of freedom in the study selects the df = 7/year, and collects the data from 2013 to 2016, so df = 28; according to the characteristics of the effects of temperature and relative humidity to the hospitalization rate of people, their degrees of freedom are all 3.

Air pollutants have a lag effect on the admission of ischemic stroke [10,11,12,13,14]. A single pollutant model was established to analyze the exposure day and lag effect. The main purpose of this study was to analyze the effect of pollutants on the number of hospitalization with ischemic stroke from lag 0 to lag 7. Lag 0 represents the exposure concentration of pollutants on the day of admission, lag 1 represents the exposure concentration of pollutants on the day before admission, and so on. In order to explore the cumulative effect of pollutant exposure, a cumulative lag effect model was established. The average lag of 1–7 days (lag 1–lag 7) was introduced into the model. The average lag of 1 day was the average of the day of admission and the previous day; the average lag of 2 days was the average of the concentration of pollutants on the admission day and the previous two days; by analogy, the relative risk (RR) of hospitalization of ischemic stroke and its 95% confidence interval were calculated when the pollutant concentration increased by 10 units under different lag days. According to different stratification, it was explored whether the effect of pollutants on hospitalization is related to age and gender. The Spearman coefficient was used to describe the correlation between various pollutants and meteorological data. The statistical analysis is completed by R 3.5.2 software (http://www.r-project.org, accessed on 11 October 2022) and mgcv data packet in the software, and the inspection level is α = 0.05.

## 3. Results

The number of hospital admissions due to cerebral infarction in Tianjin from 2013 to 2016 has been increasing year-on-year, with a total of 9081 during the study period. According to gender stratification, the number of men admitted to hospital due to cerebral infarction is more than that of women. The total population consists of 62.65% males and 37.35% females; according to age stratification, the number of residents in the 18 to 64 age group who were admitted to hospital due to ischemic stroke is less than the number of residents over 65 who were admitted to hospital due to ischemic stroke; 42.80% of residents in 18 to 64 age group and 57.19% residents over 65 (Table 1).

### 3.1. Descriptive Statistics of Air Pollutants and Meteorological Factors

During the study period, prevalence of the gaseous pollutant O_3_ showed an upward trend, while the remaining pollutants (NO_2_, SO_2_, PM_2.5_, PM_10_, and CO) showed a downward trend year-on-year. The O_3_ concentration in the meteorological data from 1 January 2013 to 31 December 2016 is the daily maximum 8 h average, and other items are 24 h averages. The meteorological data in the study period 21 October 2013–12 November 2013 and 8 April 2015–28 April 2015 were missing. The number of meteorological days included in the study was 1417 days. The study period included 1461 days of pollutant-related data. Most daily averages of SO_2_, CO and O_3_ are lower than the national secondary air quality standard. Most daily averages of NO_2_, PM_10_ and PM_2.5_ are higher than the national secondary air quality standard (Table 2).

### 3.2. Spearman’s Correlation Coefficient between Air Pollutants and Meteorological Factors

According to Spearman’s correlation coefficient results, the correlation coefficients of NO_2_, SO_2_, CO, PM_10_ and PM_2.5_ with temperature and relative humidity are negative, while O_3_ is positive (Table 3).

### 3.3. Single-Pollutant Model

When analyzing the number of hospitalized patients with ischemic stroke with a single pollutant single-day lag, the increase of NO_2_, SO_2_, CO and O_3_ has no significant effect on the number of hospital admissions due to ischemic stroke. In the single pollutant sliding average lag analysis, the increase of NO_2_, SO_2_, CO and O_3_ concentrations has no significant effect on the number of hospitalized patients with ischemic stroke.

#### 3.3.1. Associations between NO_2_ and Ischemic Stroke Hospitalizations

In the single-day lag model, when the NO_2_ concentration increased by 10 μg/m^3^, the RR, 95% confidence interval (CI) and *p* value of the number of hospitalized patients with ischemic stroke at the lag 0–7 days were analyzed. The influence of NO_2_ on the number of hospitalized patients with ischemic stroke reached the maximum at lag 0 day; the RR value was 1.007 (0.995–1.020). It can be seen that increase in NO_2_ concentration has no significant effect on the number of hospital admissions with ischemic stroke at most of the time except at lag 3 and 7(day). At lag 3 and 7(day), the RR values were 0.985 (0.973–0.997) and 0.987 (0.974–0.999), respectively (Figure 1).

#### 3.3.2. Associations between SO_2_ and Ischemic Stroke Hospitalizations

In the single-day lag model, when the SO_2_ concentration increased by 10 μg/m^3^, the RR, 95% CI and *p* value of the number of hospitalized patients with ischemic stroke at the lag 0–7 days were analyzed. The influence of SO_2_ on the number of hospitalized patients with ischemic stroke reached the maximum at lag 1 day; the RR value was 1.004 (0.994–1.014), which was not statistically significant. At lag 7(day), the RR value was 0.987 (0.977–0.998). It can be seen that increase in SO_2_ concentration has no significant effect on the number of hospital admissions with ischemic stroke most of the time (Figure 1).

#### 3.3.3. Associations between CO and Ischemic Stroke Hospitalizations

In the single-day lag model, when the CO concentration increased by 10 mg/m^3^, the RR, 95% CI and *p* value of the number of hospitalized patients with ischemic stroke at the lag 0–7 days were analyzed. The influence of CO on the number of hospitalized patients with ischemic stroke reached the maximum at lag 0 day; the RR value was 1.178 (0.851–1.630). It can be seen that increase in CO concentration has no significant effect on the number of hospital admissions with ischemic stroke at lag 0–7(day) (Figure 1).

#### 3.3.4. Associations between O_3_ and Ischemic Stroke Hospitalizations

In the single-day lag model, when the O_3_ concentration increased by 10 μg/m^3^, the RR, 95% CI and *p* value of the number of hospitalized patients with ischemic stroke at the lag 0–7 days were analyzed. The influence of O_3_ on the number of hospitalized patients with ischemic stroke reached the maximum at lag 1 day; the RR value was 1.006 (0.999–1.013). It can be seen that increase in O_3_ concentration has no significant effect on the number of hospital admissions with ischemic stroke at lag 0–7(day) (Figure 1).

## 4. Discussion

This study explored the effect of gaseous pollutant concentration on the number of hospitalized patients with ischemic stroke in Tianjin. During the study period, the air pollution situation in Tianjin gradually improved, and except for the increase of ozone concentration, concentration of all pollutants showed a downward trend. The number of hospitalized patients with cerebral infarction in Tianjin showed an increasing trend year-on-year from 2013 to 2016, especially in men and people over 65 years old. The results showed that no distinct associations between gaseous pollutants and the number of hospitalized patients with ischemic stroke most of the time. Furthermore, the effects of NO_2_, SO_2_, O_3_ and CO on ischemic stroke were not related to age and gender.

At present, more and more attention has been paid to the influence of gaseous pollutants on the incidence of ischemic stroke. The study in Guangzhou (Guo, P et al.) showed that there was a significant correlation between NO_2_ exposure and the risk of ischemic stroke [15]. The results of Lee also showed that an increase of NO_2_ concentration increased the risk of stroke [16]. Shah, AS found that the increase of hospitalization or death due to stroke is related to the increase of SO_2_ concentration [17]. Nhung, NTT in Hanoi showed that there was a strong positive correlation between stroke hospitalization rate and SO_2_ concentration [18]. Some surveys posit that there is no significant relationship between NO_2_, SO_2_ and stroke, such that Cai et al. showed that NO_2_ is not significantly associated with ischemic stroke and other cerebrovascular diseases through three cohort comparative studies [19]. Similarly, there was no association between NO_2_, SO_2_ and the number of hospitalized patients with ischemic stroke most of the time in this study, except at lag 3 and lag 7 day.

The results of this study showed that the changes of CO and O_3_ concentration had no effect on ischemic stroke. Many studies have shown that there is no significant association between CO and stroke hospitalization rate. The conclusion of this study is consistent with most studies. However, some studies have different views. A study in Taiwan pointed out that CO can increase the incidence of stroke [20]. The level of O_3_ in Tianjin is on the high side. Henrotin, JB et al. found that in the single pollutant model, the occurrence of ischemic stroke in men over 40 years old was positively correlated with the level of O_3_ 1 and 0 days behind [21]. Chen, C showed that the increase of O_3_ concentration on the same day and the first and second day (lagged) could increase the hospitalization rate of stroke [22]. Wang, Z and other studies show that low O_3_ concentration can increase the risk of cerebral infarction, and that the effect is non-linear and lagging [23]. However, the study of Nascimento, LF et al. in San Jose dos Campos, Sao Paulo, Brazil, concluded that O_3_ was not related to stroke hospitalization rate [24].

Based on the information regarding gas pollutants, meteorology and the number of hospitalized patients with ischemic stroke in Tianjin from 2013 to 2016, the generalized additive model of time series was used to control the effects of temperature, relative humidity, holidays and days of the week. The results show that the effects of gas pollutants on ischemic stroke is more objective. This study has several limitations: (1) In this study, the levels of several common air pollutants were measured–while the pollutants were not classified in more detail, according to previous research data, the main types of pollutants were included in this study; (2) The use of ambient air pollutant levels as a substitute for personal measurements is expected to lead to exposure measurement errors, and the spatial heterogeneity of each pollutant may increase the uncertainty of the results; (3) This study is a single center–it is necessary to expand the research scope, increase the sample size and jointly carry out multi-center research in order to draw more convincing conclusions in the future.

In summary, we believe that we should conduct an in-depth study to improve understanding of the relationship between air pollutants and ischemic stroke, and take this as a basis to formulate air quality standards and related policies.

## 5. Conclusions

The results of this study showed that from 2013 to 2016, the concentrations of PM_2.5_, PM_10_, CO, NO_2_ and SO_2_ in Tianjin showed a downward trend, while the concentration of O_3_ showed an upward trend, and the number of patients with ischemic stroke increased year-on-year, especially in men and people over 65 years old. Short-term exposure to air pollutants NO_2_, SO_2_, CO and O_3_ did not affect the number of hospitalized patients with ischemic stroke most of the time. Multicenter studies in the future are warranted to explore the relationships between gaseous pollution exposure and ischemic stroke.

## Figures and Tables

**Figure 1 ijerph-19-13344-f001:**
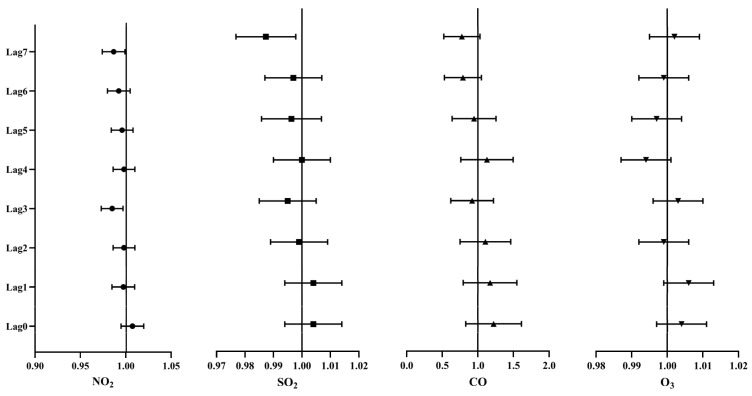
Percentage change with 95% CI of daily hospitalizations for ischemic stroke with 10 μg/m^3^ or 10 mg/m^3^ (mg/m^3^ for only CO) increase in gaseous pollutant concentration in single-day lag analysis in Tianjin (2013–2016).

**Table 1 ijerph-19-13344-t001:** Demographic Characteristics of hospitalizations for ischemic stroke in Tianjin Medical University General Hospital from 1 January 2013 to 31 December 2016.

Variables	Ischemic Stroke
Total	9081
Sex	
Male (%)	5690 (62.65)
Female (%)	3391 (37.35)
Age (Mean ± SD)	67.05 ± 11.81
18–64 (%)	3887 (42.80)
≥65 (%)	5194 (57.19)

Note: SD, standard deviation.

**Table 2 ijerph-19-13344-t002:** Descriptive statistics of air pollutants * and meteorological factors in Tianjin from 1 January 2013 to 31 December 2016.

Variables	Mean ± SD	Minimum	Frequency Distribution			Maximum
			25th	50th	75th	
Air pollutant						
CO (mg/m^3^)	1.57 ± 0.84	0.3	1.0	1.4	1.9	8.7
NO_2_ (μg/m^3^)	49.56 ± 19.00	10	34	45	61	177
O_3_ (μg/m^3^)	80.37 ± 49.37	3	42	64	117	264
PM_10_ (μg/m^3^)	125.8 ± 74.23	17	72	109	159	490
PM_2.5_ (μg/m^3^)	79.58 ± 54.74	9	41	65	102	381
SO_2_ (μg/m^3^)	39.42 ± 40.12	3	14	26	48	280
Meteorological factor						
Temperature (°C)	13.96 ± 11.29	−14.14	3.00	15.6	24.44	33.59
Relative humidity (%)	56.98 ± 17.82	10.86	43.43	57.43	70.29	97.00

Note: SD, standard deviation. ** According to the Chinese Ambient Air Quality Standards GB3095-2012 (CAAQS-GB3095-2012), it is recommended to take the short-term (24-h) AQG level of CO, NO_2_, PM_10_, PM_2.5_ and SO_2_ as the living area target, and it is recommended to adopt 4 mg/m^3^, 80 μg/m^3^, 150 μg/m^3^, 75 μg/m^3^, 150 μg/m^3^, respectively. The average of daily maximum 8-h mean concentrations of 160 μg/m^3^ is represented ozone’s standard.*

**Table 3 ijerph-19-13344-t003:** Spearman correlations among air pollutants and meteorological factors in Tianjin, 2013–2016.

Air Pollutant and Meteorological Factor	NO_2_	O_3_	PM_10_	PM_2.5_	SO_2_	Temperature	Relative Humidity
CO	0.78 **	−0.40 **	0.67 **	0.72 **	0.74 **	−0.39 **	−0.18 **
NO_2_		−0.47 **	0.66 **	0.67 **	0.76 **	−0.46 **	−0.19 **
O_3_			−0.18 **	−0.18 **	−0.47 **	0.76 **	0.22 **
PM_10_				0.91 **	0.58 **	−0.19 **	−0.11 **
PM_2.5_					0.54 **	−0.20 **	−0.04
SO_2_						−0.55 **	−0.26 **
Temperature							0.20 **

Note ** *p* < 0.01.

## Data Availability

The meteorological data come from China Meteorological data Service Network (http://data.cma.cn, accessed on 11 October 2022), The air pollutant data come from Tianjin Eco-Environmental Monitoring Center (http://www.tjemc.org.cn, accessed on 11 October 2022).

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
