# Peer review of "Acute Gaseous Air Pollution Exposure and Hospitalizations for Acute Ischemic Stroke: A Time-Series Analysis in Tianjin, China"

_ijerph, 2022, doi:10.3390/ijerph192013344_

Round 1

Reviewer 1 Report

Major comments

1.       Line 138-199 (Results from time-series regression): The underlying sentence need to be addressed.

-          Please report the effect estimate using RR (95%CI: lower CI, higher CI) throughout the manuscript. The author does not need to report p-value after illustrating the interval ranges.

-          Overall, null associations between acute ischemic stroke and air pollution were observed. Effect estimates (RR) were not significant different for all lags.

-          Negative associations were observed for some lags, this can be occurred by chance. Moreover, no significant different between the RR in each lag. Thus, the results should be described and concluded more general.

-          The effect estimates are more likely to be large in the immediate lags (lag0 and lag1).

-          Please do not use the report pattern (duplicate the sentence of different air pollutants).

-          Figure 1-4 and Table 4: The information in figure 1 and table 4 are similar. It would be better to move table 4 to supplementary material and keep figure 1 in the maintext.

-          Figure 1-4: Please combine Figure 1-4 into one figure and add the caption to describe each panel results.

2.       Line 186: Why the author performed two-pollution model as the findings revealed null-association between air pollution and health outcomes in the single-pollution model.

Minor comments

1.      Abstract: The author should include the introduction in the abstract.

2.      Abstract: Change the overall summary to be more general as the findings of this results presented a null association (negative association were found in some lags and the effect estimates between each lag were not different as the 95%CI is overlapping).

3.      Line 40-42: “Some foreign articles have 40 shown that exposure to air pollutants such as NO2, SO2, CO and O3 increases the risk of 41 cerebrovascular disease, hospitalization and even death.” Please cite more references.

4.      Line 50-52: “Therefore, the purpose of this study is to explore the effects of short-term exposure of gaseous pollutants NO2, SO2, CO and O3 on the number of patients with cerebral infarction in Tianjin.” As it was mentioned in the previous sentence that air pollution in this area is polluted, why the author did not include PM in this study?

5.      Line 59-61: “The patient information comes from the ID card registration and medical insurance related information when the patient is admitted to the hospital, and the Tianjin residents who have been over 18 years old are screened out” Please correct this sentence as the subjects of this study are the patients age over 18 years.

6.      Line 62-64: “The diagnosis of the patient is based on the discharge diagnosis, referring to the International Classification of Diseases (ICD-10), and the study selects the disease as cerebral infarction (I63.900)” Why author did not include all types of acute ischemic stroke in the analyses (ICD10: I63.0-I63.9)?

7.      Line 64-74: Please make another sub-title namely “2.2 Environmental data”.

8.      Line 65-66: “including daily average temperature (temperature,temp) and relative humidity ((relative humidity, rh)).” Please add the unit of temperature and relative humidity.

9.      Line 70-74: I would think this sentence is not necessary as the author can replace this sentence with “ground monitoring station”.

10.  Line 85: ““α”is the residual” Please correct “residual” to “intercept”.

11.  Statistic method section: Please describe how to perform descriptive statistic test (correlation test) for air pollution in this study.

12.  Statistic method section: Please describe how the model selection were performed to obtain the “minimal adequate model” in this study.

13.  Line 94: “Air pollutants have lag effect on the admission of cerebral infarction.” Please cite this sentence.

14.  Line 99-101 and 106-108: “Through the established model, the relative risk of hospitalization of cerebral infarction and its 95% confidence interval were calculated when the pollutant concentration increased by 10 units under different lag days.” These sentences were duplicated in the methods section, please shorten it.

15.  Line 110-111: “In addition, single-pollutant model and two-pollutant model are used to evaluate the stability of the model.” Please describe how two-pollutants models were performed in this study.

16.  Line 118 and 122: “The ratio is about 1.678:1 (5690:3391) and The proportion is about 1:1.336 (3887:5194)” Please consistently illustrate the result using percent (%) as it was presented in the Table 2.

17.  Line 124-125: “During the study period, the gaseous pollutant O3 showed an upward trend, while the remaining pollutants NO2, SO2, PM2.5, PM10, and CO showed a downward trend year by year.” Please provide the time series plots of each pollutant (this can be presented in the supplementary files) to show the trend of the air pollution.

18.  Line 129-130: “The missing data were supplemented by interpolation method.” Please move this sentence into the methods part (meteorological data).

19.  Line 131-133: “The annual averages of SO2, CO and O3 are all lower than the national secondary air quality standard. The annual averages of NO2, PM10 and PM2.5 are higher than the national secondary air quality standard (Table 2).” The author should provide daily average concentration of the air pollutions as the objective of this study focused for short-term (daily) effect of air pollution in health outcomes.

20.  Results: The author should illustrate the correlation test (Table 3) in the result section.

21.  Line 186: “Select lag 4 day data to establish a multi-pollutant model, and introduce NO2, SO2, O3, PM2.5, PM10 into the CO model to establish a two-pollutant model.” This sentence should be moved to methods part (related to #Minor comments 15). Moreover, why lag 4 was selected because the large effect estimates are mostly found in the immediate lags (lag 0 or lag 1)

22.  Line 206-207: The author should emphasize how the level of air pollution (compared to national air pollution standard/ or guideline).

Author Response

We are pleased to get your comments and suggestions.We have made corresponding adjustments and look forward to your suggestions.

We have doubts about the first major comment of Reviewer 1. Could you give us further suggestions or model articles for reference? We look forward to hearing your thoughts and making changes.Thank you for your help.

QUESTION:Why the author performed two-pollution model as the findings revealed null-association between air pollution and health outcomes in the single-pollution model.

ANSWER:In the single-pollutant model, different pollutants are introduced to establish a multi-pollutant model to judge whether the established model is reliable, and to explore whether the impact of different pollutants on ischemic stroke is affected by other pollutants.We want to explore whether the impact of pollutants on ischemic stroke can be superimposed or offset.

QUESTION:“Therefore, the purpose of this study is to explore the effects of short-term exposure of gaseous pollutants NO2, SO2, CO and O3 on the number of patients with cerebral infarction in Tianjin.” As it was mentioned in the previous sentence that air pollution in this area is polluted, why the author did not include PM in this study?

ANSWER:Because the PM part is participating in another study.

QUESTION:why lag 4 was selected because the large effect estimates are mostly found in the immediate lags (lag 0 or lag 1)

ANSWER:We have observed better results in lag 4 , and the specific mechanism remains to be explored.

The supplement is at the end of the article

Reviewer 2 Report

I'd like to thank you for giving me this opportunity to review this study. Although the topic of this study is of much interest, there are several issues need ti be addressed.

First, the introduction section needs to improve. It is suggested to discuss more about health burden attributable to ambient air pollution. Please consider the following prestigious papers.

https://ehjournal.biomedcentral.com/articles/10.1186/s12940-022-00832-4

For the section of methods, as the authors might know, one of the most important sections in this type of studies is air quality data processing and validation. The authors must report the source of data, hourly data coverage, the approach of data validation. It is highly recommended that authors re-organized the method section. To do this, please apply the following papers.

https://www.nature.com/articles/s41598-019-56578-6

Author Response

We are pleased to get your Comments and Suggestions.We have made corresponding adjustments and look forward to your suggestions.

Round 2

Reviewer 1 Report

1. According to the main results (Figure 1), generally, no associations between gaseous pollutants and ischemic stroke were observed in this study. However, the author stated that “Short-term exposure to air pollutants NO2 and SO2 reduced the number of hospitalized patients with ischemic stroke. in the conclusion” Is this means exposure to NO2 and SO2 is a protective factor to reduce the risk of ischemic stroke? This can lead to misinterpret of the overall results. My suggestion is that the results and conclusion should be more generally illustrated (e.g., using lag 0 to present the results and summarize as no associations were observed among all pollutants in this study).  

2. From the author’s response “We have observed better results in lag 4 , and the specific mechanism remains to be explored.”, what the mean of “better results”? According to previous study[1] and the results in this study, immediate lag should be selected to further investigate co-pollutant effects.   

[1] Huang F, Luo Y, Tan P, Xu Q, Tao L, Guo J, Zhang F, Xie X, Guo X. Gaseous Air Pollution and the Risk for Stroke Admissions: A Case-Crossover Study in Beijing, China. Int J Environ Res Public Health. 2017 Feb 14;14(2):189. doi: 10.3390/ijerph14020189. PMID: 28216595; PMCID: PMC5334743.      

Author Response

Dear Reviewer:

Thank you for your suggestions and reminders.

For comment 1, we have explained the results and conclusions more comprehensively. Using the immediate lags to present the results and summarize. All parts related to this main idea have been corrected.

For comment 2, I sincerely apologize for the inappropriate explanation of using “better results” in the response. I'm sorry that our mistakes and lack of experience have wasted your valuable time and energy. As the findings revealed null-association between air pollution and health outcomes in the single-pollution model and multi-pollution model. We follow the suggestion in the review report (round 1) to delete relevant parts of the multi-pollutant model.

We will pay more attention to this manuscript and be ready to reply to any questions you may have.

We look forward to hearing from you soon.

Have a nice day.

Best regards.

Reviewer 2 Report

the paper is ready for publication.

Author Response

Dear Reviewer:

Thank you for your consideration of this manuscript.

Have a nice day.

Best regards.